# Barriers and enablers to young people accessing sexual and reproductive health services in Pacific Island Countries and Territories: A scoping review

Maggie Ikinue Baigry[1]*, Robin Ray[2], Daniel Lindsay[3,4], Angela Kelly-Hanku[5,6], Michelle Redman-MacLaren[1]

1 College of Medicine and Dentistry, James Cook University, Cairns, Queensland, Australia, 2 College of Medicine and Dentistry, Anton Breinl Research Centre for Health Systems Strengthening, James Cook University, Townsville, Queensland, Australia, 3 QIMR Berghofer Medical Research Institute, Brisbane Queensland, Australia, 4 School of Public Health, University of Queensland, Brisbane, Queensland, Australia, 5 Sexual and Reproductive Health Unit, Papua New Guinea Institute of Medical Research, Goroka, Eastern Highlands Province, Papua New Guinea, 6 Global Health Equity and Justice Research Group, Kirby Institute, University of New South Wales, Sydney, New South Wales, Australia

* maggie.baigry@my.jcu.edu.au

**Data Availability Statement:** All relevant data are within the manuscript and its Supporting Information files.

## Abstract

### Background

The number of young people utilising sexual and reproductive health services in Pacific Island Countries and Territories remains poor despite the availability and the existence of the fundamental rights to access these services. Adolescents and youth need accurate information and timely access to contraceptives to prevent adverse consequences associated with unintended pregnancies, abortion, childbirth and untreated sexually transmitted infections. This scoping review identifies and analyses factors contributing to young people's low access to sexual and reproductive health information and services in this region.

### Methods

Guided by the PRISMA Scoping review guidelines, we searched three databases (Medline Ovid, Scopus and CINAHL Complete) for peer-reviewed articles published between 1st January 2000 and 31st August 2020 that reported on factors, including barriers and enablers, affecting access to sexual and reproductive health information and services by young people living in Pacific Island Countries and Territories. We assessed the quality of each study according to the study designs, methods of data collection, data analysis and ethical considerations. All information was sorted and organised using an Excel Spreadsheet. Text data from published articles were charted inductively using thematic analysis with no predetermined codes and themes.

### Findings

Five hundred eighty-nine articles were screened, and only eight met the inclusion criteria outlined in this scoping review protocol. These eight articles reported studies conducted in

**Funding:** The authors received no specific funding for this work.

**Competing interests:** The authors have declared that no competing interests exist.

four Pacific Island Countries and Territories: Cook Islands, Fiji, Papua New Guinea, and Vanuatu. Factors such as lack of accurate sexual and reproductive health knowledge and social stigma were the leading causes of young people's limited access to sexual and reproductive health services. Cultural and religious beliefs also invoked stigmatising behaviours in some family and community members.

## Conclusion

This scoping review revealed that social stigma and judgemental attitudes imposed by family and community members, including healthcare providers, hinder young unmarried individuals in Pacific Island Countries and Territories from accessing sexual and reproductive health information and contraceptives. Alternatively, a non-judgmental healthcare provider is perceived as an enabler in accessing sexual and reproductive health information and services. Moreover, given that only a few studies have actually focused on young people's sexual and reproductive health needs in the region, more research is required to fully understand the health-seeking behaviours of young people in their specific contexts.

## Introduction

Timely access to sexual and reproductive health (SRH) information and services by young adults in Pacific Island Countries and Territories (PICTs) remain low despite the availability of SRH services [1]. Globally, young adulthood is a critical time of development where many opportunities and challenges exist. One such challenge is access to healthcare services [2]. For instance, The Society of Adolescent Health and Medicine has identified from over 40 publications in the United States that young adults between 18 and 25 years of age experience higher rates of mortality and unintended pregnancies and they have lower access to healthcare services compared with those immediately younger (10–17 years of age) and those immediately older (26–30 years of age) [2]. Moreover, with unique sexual health needs, young adults need accurate information and timely access to contraceptives to prevent adverse consequences of unintended pregnancies, abortion, childbirth and untreated sexually transmitted infections (STIs) [3–6].

In the Pacific region, multiple factors determine people's access to SRH information and services, including culture, economic and geographical location [7]. For example, a report from Papua New Guinea that aimed at better understanding the country's allocation and spending of public funds found a shortage of healthcare providers, delay or lack of funding, including payments of healthcare providers, and limited supply of drugs [8] have hindered service delivery and accessibilities. People most likely to experience difficulties accessing SRH services are those living in rural and remote areas and populations with specific needs, such as young people, persons with disabilities and people living with human immunodeficiency virus HIV and people identified as lesbian, gay, bisexual, transgender, queer and intersex (LGBTQI) [7]. Consequently, the global effects to ensure SRH rights to quality and accurate SRH information and services are undermined [9], resulting in instances such as low contraceptive prevalence rate and high unmet need for contraceptives among young sexually experienced women in some PICTs [10].

Unintended pregnancies and STIs are common among young women [11]. In five PICTs (Papua New Guinea, Solomon Islands, Vanuatu, Nauru and the Marshall Islands), young

women's fertility rates remain high, at over 50 live births per 1,000 women aged 15–19 years [10]. It is widely acknowledged that many unintended pregnancies are terminated using unsafe abortion practices [7], as evidenced in studies from Papua New Guinea [12, 13]. The most prevalent STI among young women in the Pacific region is *Chlamydia trachomatis*. A study of 1,618 pregnant women in six PICTs (Fiji, Kiribati, Samoa, Solomon Islands, Tonga, and Vanuatu) between 2004 and 2005 showed that 26.1% and 11.9% of women under 25 years and women aged 25 years, respectively, were infected with *Chlamydia trachomatis* [14]. In Fiji, a recent study showed that 38.8% of young women aged 18–24 years were infected with *Chlamydia trachomatis* [15], while in the Solomon Islands, the prevalence of *Chlamydia trachomatis* was reported as 20% among women aged 16–49 years attending female clinics in Honiara [16].

Several studies have previously explored young people's sexual risk behaviours in PICTs [17–19], including the use of condoms and contraceptives [20–22]. Some studies have explored young people's SRH issues related to HIV testing, treatment and prevention [23–27]. Other studies have explored access to family planning services and antenatal clinics [28]. However, to the authors' knowledge, there has been no systematic review of the literature to determine the perceived barriers and enablers of young adults accessing SRH information and services in the PICTs. Thus, this scoping review seeks to answer two main questions:

1. What has been reported about the SRH of young people in PICTs?

2. What has been reported on young people's perception and practices of accessing and using sexual and reproductive health services?

## Methods

### Protocol

A scoping review was conducted following the guidelines for the Preferred Reporting Items for Systematic reviews and Meta-Analysis extension for Scoping Review (PRISMA-ScR) [29]. With the increase in the number of published scoping reviews and the lack of consistency in the methodology and reporting of results, the PRISMA-ScR checklist was developed following the guidelines of Levac, Colquhoun and O'Brien [30], who built upon the scoping review methodology of Arksey and O'Malley [31]. A protocol was written (unpublished; S1 Appendix) to guide the search for both qualitative and quantitative studies that would help answer our review questions. Medline (Ovid), CINAHL Complete and Scopus databases were searched using keywords and medical subject headings (MeSH) terms between 7th and 18th September 2020 (S2 Appendix). These three databases have a comprehensive overview of global literature in the fields relating to human life. We also searched Google Scholar, PubMed, -JCU One Search and Web of Science for additional information.

### Eligibility criteria

Eligibility criteria are as follows;

1. Peer-reviewed articles that explore young people's responses concerning sexual and reproductive health information and services.

2. Articles with young people aged 10–24 years as the main study participants living in one of the PICTs, namely American Samoa, Cook Islands, Federated States of Micronesia, Fiji, French Polynesia, Guam, Kiribati, Marshall Islands, Nauru, New Caledonia, Niue, Northern Mariana Islands, Palau, Papua New Guinea, Pitcairn Islands, Samoa, Solomon Islands,

Tokelau, Tonga, Tuvalu, Vanuatu, Wallis and Futuna. Studies of young Pacific Islander adolescents and youth living outside PICTs were not considered based on socio-economic and cultural differences.

We limited our search to include articles published in English between 1[st] January 2000 and 31[st] August 2020. We were interested to understand what had transpired in the Pacific region during these twenty years, as there had been targeted SRH service activity linked to the Millennium Development Goals (MDG) and, more recently, the Sustainable Development Goals (SDG). In 2000, the United Nations Member States embarked on eight MDGs to fight poverty and combat issues hampering human development progress [9]. In these MDGs, goal five aimed to improve maternal health through two indicators; reduction of maternal mortality by three quarters and access to reproductive services.

## Data screening and charting process

The literature identified using keywords and MeSH terms were downloaded into an Endnote library (Version 9) and screened using the established inclusion and exclusion criteria (S1 Appendix). After the screening process, we assessed the quality of each peer-reviewed article in our inclusion list according to the types of study design, methods involved in data collection and analysis and ethical considerations [32]. We then extracted data from the included studies using a standardised template in an Excel spreadsheet and organised text data according to; the age and gender of the participants, location of the study, study design, method of data collection, ethical considerations and SRH issues such as access to contraceptives and STI detection and treatment. Our text data were analysed inductively using thematic analysis [33]. The results sections of each included article were read several times for familiarisation with no predetermined codes. Finally, codes were created using NVivo (12 Plus version) software and linked together using thematic maps [33] to determine the themes related to barriers and enablers to young people's access to SRH information and services in PICTs.

## Results

### Identification of records

Titles and abstracts of 1265 articles were screened by the lead investigator (MB), resulting in the exclusion of 1230 articles, with 35 articles considered eligible for full-text screening. After the full-text screening by four authors (MB, MRM, RR and DL), 27 articles were removed, and the remaining eight were included for evaluation in this scoping review (See Fig 1). These are the reasons for the removal of the 1257 articles. Our search had 676 articles that were duplicates; 474 articles had titles with study settings outside of PICTs; 80 articles did not contain information about access to SRH information and services; 13 articles did not have young people aged 10–24 years as study participants, and 14 articles did not specifically report access to SRH services in PICTs.

### Characteristics of sources of evidence

The eight articles [34–41] included for evaluation were studies conducted in PICTs with young people aged 10–24 years as main study participants and reported on young people's responses to access to SRH information and services, thus meeting the inclusion criteria for this scoping review. Table 1 presents a summary of the studies' characteristics. All eight articles employed a qualitative design, and two [34, 35] were from studies conducted in Papua New Guinea. Two articles [38, 39] were from studies conducted in Fiji, two articles [36, 37] were

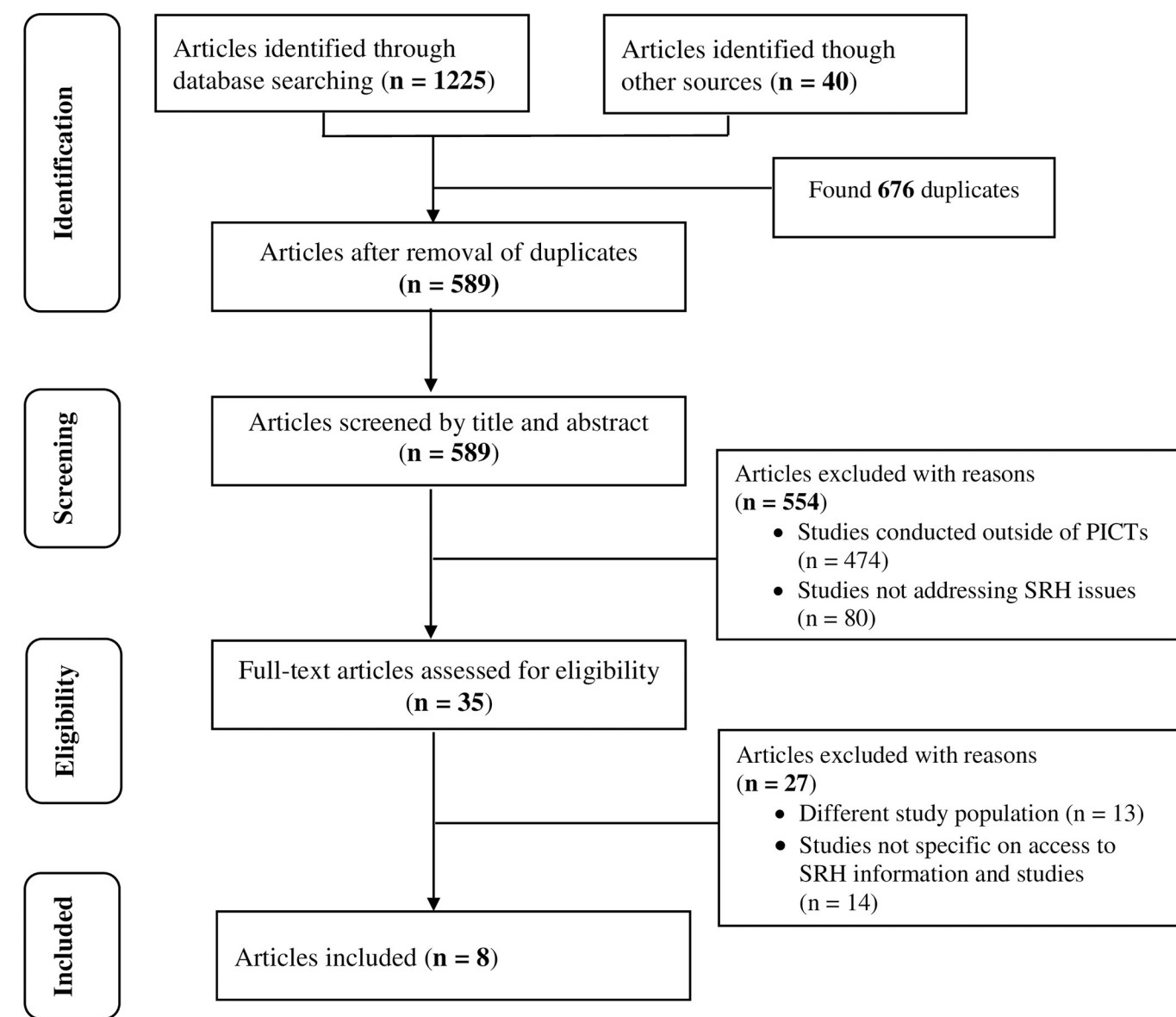

**Fig 1. PRISMA-ScR flow diagram of the scoping review process.**

from one study conducted in Vanuatu, and the final two articles [40, 41] were from one study conducted in the Cook Islands.

Four [36–39] of the eight articles reported data collected using focus group discussions and semi-structured interviews. In three articles [35, 40, 41], data were collected using only semi-structured interviews. The remaining article [34] reported observation and storytelling as data collection methods. Study participants for six articles [34–39] were males and females, while two articles [40, 41] had females only as study participants.

Two articles reported on young people's barriers and enablers to accessing SRH information and services [36, 37]. The other six articles reported some elements related to SRH information and services access. Two articles [34, 40] explored issues of contraceptive use, including access to contraceptive information and services, while two articles [35, 39] reported

**Table 1. Summary of the included articles.**

| Citation | Study setting | Type | Study design | Method of data collection | Age range in years (10–24; ≥ 80% under 24) | Sex | Focus of the study |
|---|---|---|---|---|---|---|---|
| Hemer (2019) | Papua New Guinea | Peer-reviewed | Qualitative | The author's lived experience and observations | 15–25 | Male | Contraceptive advise and family planning services in an isolated community |
| | | | | | | Female | |
| Keck (2007) | Papua New Guinea | Peer-reviewed | Qualitative | Semi-structured interviews | 15–30 | Male | HIV/AIDs knowledge, access, and usage of condoms among a young rural, remote population. |
| | | | | Author's field notes and observations | | Female | |
| Kennedy et al. (2013) | Vanuatu | Peer-reviewed | Qualitative | Focus groups discussion and semi-structured interviews | 15–19 | Male | Perceptions of youth-friendly sexual reproductive health services |
| | | | | | | Female | |
| Kennedy et al. (2014) | Vanuatu | Peer-reviewed | Qualitative | Focus groups discussion and semi-structured interviews | 15–19 | Male | Adolescent's excess sexual and reproductive health information |
| | | | | | | Female | |
| Mitchell & Bell (2020) | Fiji | Peer-reviewed | Qualitative | Focus group discussion, interviews | 18–29 | Male | Understandings of premarital sex and condom use among young people in Fiji |
| | | | | | | Female | |
| O'Connor et al. (2019) | Fiji | Peer-reviewed | Qualitative | Focus group discussion and key informant interviews | 15–19 | Male | Adolescents' emotional experiences and health-seeking behaviours in Fiji |
| | | | | | | Female | |
| White et al. (2018a) | Cook Islands | Peer-reviewed | Qualitative | Individual in-depth interview | 14–19 | Female | Contraceptive knowledge, attitude and use among young women in the Cook Islands |
| White et al. (2018b) | Cook Islands | Peer-reviewed | Qualitative | Individual in-depth interview | 14–19 | Female | Narratives of young women who were pregnant before the age of 20 years. |

on HIV knowledge and explored condom use in premarital sex. In addition, two articles [39, 41] reported on the experiences of young people regarding health-seeking behaviour for SRH information and services.

## Young people's barriers to accessing sexual and reproductive health services

**Lack of sexual and reproductive health knowledge.** Young people reported not accessing SRH services because they were not aware of such services [37]. For example, information about sex, including SRH services, was not always available to young unmarried men and women in many communities in PICTs because parents believed young unmarried people were too young to receive such information [36, 37]. Thus, young people also perceived that they were too young to access SRH services [36]. Young people believed SRH services, especially family planning services, were for married men and women and were unavailable to them [36, 40]. In addition, the actions of healthcare providers giving condoms and contraceptives to only married men and women affirmed the perception that only married people were eligible to access family planning services [35, 36].

A lack of understanding about puberty and reproduction has created fear in young people about accessing SRH services. Young men and women stated that not knowing what to ask for or discuss with a healthcare provider was a reason for not accessing a SRH service [37]. As a result, many unmarried individuals have not attended a SRH service [36, 41]. In addition, reports from their friends or other young people about being scolded or lectured by healthcare providers further discouraged their access to a SRH service [36].

**Cultural and religious practices.** Lack of SRH knowledge and misconceived ideas about the purpose of SRH services also stemmed from cultural communication practices that restricted intergenerational dialogue between young people, parents, and other adults about sex or sexual health education [38]. The notion that premarital sex is taboo and teaching

young men and women about sex would encourage them to engage in premarital sex is the basis for which parents, older relatives and other members of communities were reportedly uncomfortable discussing sexual issues with young, unmarried individuals [35, 38]. Moreover, young individuals and healthcare providers were both reluctant to discuss SRH problems [36, 37]. Young men and women expressed anxiety when discussing sexual matters and were worried about physical examinations [36, 37]. Healthcare providers, who are in the position to provide advice and treatment, are uncomfortable discussing sex-related matters with unmarried young people [36, 37, 39].

**Attitudes of healthcare providers.** Many young unmarried men and women described healthcare providers as having a judgmental attitude toward them [35–41]. In nearly all eight studies, participants feared healthcare providers lecturing and scolding them publicly [34–39]. In addition, participants did not trust healthcare providers to maintain their privacy, as healthcare providers were among some community members involved in orchestrating their public shame and humiliation [35–38, 40, 41]. Young people in rural communities feared healthcare providers as some were their relatives and neighbours [40, 41].

**Location of sexual reproductive health facilities.** The location of SRH facilities, such as hospitals and clinics, impacts accessibility. Young, unmarried people mentioned they preferred accessing SRH facilities that were not within their communities to avoid the judgmental attitudes of people they know [39]. For example, rural adolescents considered urban areas to offer greater anonymity [39]. Furthermore, in many rural communities, there are limited healthcare infrastructures. For instance, there is often only one healthcare facility for all healthcare needs [34, 35]. This resulted in a lack of privacy for young people needing confidential SRH services. Young people mentioned the lack of privacy at healthcare facilities has made them reluctant to access services offered [35–41].

For those young unmarried people who did access SRH services, the cost of travel and service fees were additional barriers [40, 41]. Many young adults in PICTs relied on their parents and families for financial support and could not afford fees. These young people recommended that services and commodities such as condoms and contraceptives be freely accessible for health and well-being [36, 37].

**Keep the premarital sexual activities hidden.** Young, unmarried men and women in many PICTs kept their sexual relationships hidden from their parents, older relatives, and other members of their communities for fear of gossip, public shame, and embarrassment [34, 36–41]. Young men and women have stated that when people in their communities have discovered or suspected that they have had sex, people (especially their friends and peers) tease and spread gossip about their sexual behaviour [37–39]. Gossiping and teasing tarnish the reputation of young people in the community and reduce their chance of having a good married life [38, 39]. Young men and women were also afraid that exposure of their premarital sexual behaviour would disgrace themselves and their families [35–41]. For these reasons, young unmarried people in the Pacific were not accessing SRH services because they did not want other people to know that they were sexually active [36, 37, 39, 40]. To be seen at a family planning or a STI clinic is an indirect, non-verbal expression understood by the people in their communities to indicate that they have engaged in sex [41].

## Enablers of access to sexual and reproductive health services for young people

**Increase sexual and reproductive health knowledge.** To overcome the hurdles associated with opposition and disapproval for young people accessing SRH information and services, young men and women in PICTs reported that greater awareness was needed to increase

parents' SRH knowledge [36]. Parents' lack of sexual health education was a reason for poor parent-adolescent communication, opposition and disproval of accessing SRH services for counselling, commodities, and STI treatment [36, 37, 39]. Workshops using drama that target parents, community elders, and leaders may increase sexual health knowledge in the communities [36, 37, 40] In addition, some young unmarried people reported that some mothers are supportive [38, 40, 41]; however, they need accurate sexual health information.

Many young men and women mentioned that lack of SRH knowledge and fear of not knowing what to ask for or discuss with healthcare providers was why they did not seek access to a SRH service when needed. To overcome this hurdle, young people have proposed ways to increase their own SRH knowledge. One way was for peer educators and nurses to visit schools and share sexual health information [37]. Another way was for teachers to teach the sexuality education curriculum at primary and secondary schools [37]. Young people describe nurses as the trusted source to deliver SRH information [35–37] because they are trained professionals who can provide advice and treatment [37]. Other ways to increase SRH knowledge included posters, comics, pamphlets, and radio programs [34, 35]. Young people reported they also obtained SRH information from Facebook [37].

**Encouragement and support from parents and friends.** Young men and women's personal barriers of fear, shame and embarrassment were reasons for not seeking access to a SRH services. However, encouragement, emotional and financial support from friends, parents and community members made it easier to seek and access SRH services [36, 40].

**Friendly, non-judgmental, and kind healthcare providers.** Young unmarried people described their preferred feature of a SRH service as that of a healthcare provider who is friendly, non-judgmental, and kind [36, 37]. A professionally trained person who understands young people's SRH rights and maintains patient confidentiality [36, 37]. In addition, a standalone SRH clinic is desirable [36, 37].

## Discussion

This scoping review sought to understand young unmarried people's challenges when seeking access to SRH information and services. Specifically, we wanted to know: 1) what has been reported about SRH of young people in PICTs; 2) what was reported on young people's perception and practices of accessing and using SRH services? To answer these questions, we conducted a systematic scoping literature review of relevant databases, identified relevant studies, and examined young people's responses published in peer-reviewed articles.

Barriers to accessing SRH information and services among young people included a lack of SRH knowledge and social stigma associated with premarital sexual practices. Social stigma in the context of SRH is disapproval of or discrimination against individuals or groups based on social norms and affects the recipients of SRH services [42]. Findings showed that young person's fear, shame, and embarrassment were brought on by the judgmental attitudes of healthcare providers, parents, and members of their communities. Sexual and reproductive health information were not always communicated to young unmarried men and women in PICTs because of restrictive cultural communication practices. Thus, growing up, adolescents and youths had limited access to accurate SRH information within family units and had misconceived ideas of SRH services [43].

These findings are consistent with the Secretariat of the Pacific Community's report on barriers and enablers to SRH services in the Pacific region [7], along with other studies conducted in the Pacific region and similar settings. For example, a study in Tonga shows that young people feared shame and embarrassment associated with possessing items such as condoms or being seen in venues such as family planning clinics [22]. In Kenya and Nigeria, young

unmarried people also suffer the same burden of social stigma as recipients of SRH services [44, 45]. Similarly, young people in Malaysia, Nepal, and Iran have expressed fear and shame as barriers to accessing SRH services [46–48]. Socio-cultural norms were considered deterrents to young people's access to contraceptives and STI treatments [22, 49].

Currently, young men and women in PICTs lack the resources to help them avoid unintended pregnancies and the acquisition of STIs. Evidence shows that this region's high rate of adolescent pregnancies and STIs stems from failures to address the young people's access to accurate information and quality healthcare services [7]. In addition, youth-friendly services are still lacking in rural and remote communities and on outer islands in many PICTs [7, 10, 11]. At the same time, healthcare providers' judgemental attitudes are rife toward providing emergency contraceptives and confidential counselling [10, 11] to sexually experienced adolescents and young adults.

The International Conference on Population and Development, held in Cairo in 1994, recognised SRH rights as a cornerstone for population and development programs [50]. Despite this commitment, we found that young unmarried people's SRH rights for universal access to SRH service are not fully recognised in many PICTs [51]. Insufficient and inadequate laws, policies, and guidelines may have contributed to discrimination and prevented access to SRH services [7]. Our findings indicate the urgent need for tailored SRH services that young unmarried people may easily access [52]. A tailored sexual health service may assist in achieving the sustainable development goal 3.7; *Ensure universal access to SRH care services, including family planning, information, and education, by 2030* [53]. Some PICTs have developed SRH policies [54, 55]. However, implementing these policies was problematic for various in-country reasons, such as the shortage of healthcare providers, lack of political will and resource constraints [8]. The findings in this scoping review are crucial to informing some of these policies for better, more impactful implementation, such as supporting user-friendly healthcare services.

In PICTs, more research is required to understand the drivers of young adults' health-seeking behaviour, access to SRH information and services, types of SRH services accessed, and unmet need for services [1]. Evidence from this scoping review shows that there are limited data available on the health-seeking behaviour of young adults in PICTs, perhaps because of the sensitivities around socio-cultural expectation and premarital sexual behaviour [4, 10]. In the present age of widespread use of information technology and mass movement of people away from their local Indigenous communities, further research into the SRH issues affecting young adults' quality of life must be carried out. For example, more research is required in Papua New Guinea to understand the drivers of risky sexual behaviour and the context in which unintended pregnancies occur among young people [56].

This scoping review has limitations. Firstly, we limited our search to those studies published in English, which may have limited numbers of articles retrieved from our search. Secondly, we searched only peer-reviewed articles and did not include grey literature, which may have precluded some data reported from young people in PICTs. Nevertheless, we sought to include the highest quality of evidence available to answer our review questions.

## Conclusion

This scoping review found that social stigma and judgemental attitudes imposed by family and community members, including healthcare providers, hinder young unmarried individuals in PICTs from accessing SRH information and contraceptives. However, a non-judgmental healthcare provider can be an enabler in accessing SRH information and services. Therefore, changing policies for a tailored, user-friendly SRH service will enhance access to SRH service

and help reduce adolescent pregnancies and STIs. In addition, only a few studies have focused on young people's SRH needs in the region. Therefore, more research is required to describe context-specific health-seeking behaviours of adolescents and youths in each PICTs to guide policy change and implementation.

## Supporting information

**S1 Appendix. Review protocol.**
(DOCX)

**S2 Appendix. Example of database search.**
(DOCX)

**S3 Appendix. PRISMA-ScR checklist.**
(DOCX)

**S4 Appendix. 16 items for screening.**
(DOCX)

**S5 Appendix. Keywords.**
(DOCX)

## Acknowledgments

The authors would like to thank Dr Karen Cheer for her support in reviewing the search strategy (review protocol) and the first draft of this article.

## Author Contributions

**Conceptualization:** Maggie Ikinue Baigry, Michelle Redman-MacLaren.

**Formal analysis:** Maggie Ikinue Baigry.

**Investigation:** Maggie Ikinue Baigry.

**Methodology:** Maggie Ikinue Baigry, Michelle Redman-MacLaren.

**Supervision:** Robin Ray, Daniel Lindsay, Angela Kelly-Hanku, Michelle Redman-MacLaren.

**Visualization:** Michelle Redman-MacLaren.

**Writing – original draft:** Maggie Ikinue Baigry.

**Writing – review & editing:** Maggie Ikinue Baigry, Robin Ray, Daniel Lindsay, Angela Kelly-Hanku, Michelle Redman-MacLaren.

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
