## [Decision Letter · Decision Letter 0]

31 May 2022

PONE-D-21-11951Barriers and enablers to young people accessing sexual and reproductive health services in Pacific Island Countries and Territories: a scoping reviewPLOS ONE

Dear Dr. Baigry,

Thank you for submitting your manuscript to PLOS ONE. After careful consideration, we feel that it has merit but does not fully meet PLOS ONE’s publication criteria as it currently stands. Therefore, we invite you to submit a revised version of the manuscript that addresses the points raised during the review process.

Please note that we have only been able to secure a single reviewer to assess your manuscript. We are issuing a decision on your manuscript at this point to prevent further delays in the evaluation of your manuscript. Please be aware that the editor who handles your revised manuscript might find it necessary to invite additional reviewers to assess this work once the revised manuscript is submitted. However, we will aim to proceed on the basis of this single review if possible.  Your manuscript has been assessed by an expert reviewer, whose comments are appended below. To ensure your study is up-to-date, and to address point 1 made by the reviewer, please also extend your search to include the latest studies from 2021 and 2022 if any meet your inclusion criteria. Please also address the reviewer's other points carefully in your response to reviewers, and update your manuscript accordingly.

We look forward to receiving your revised manuscript.

Kind regards,

Joseph Donlan

Editorial Office

PLOS ONE

Journal Requirements:

4. Please include your tables as part of your main manuscript and remove the individual files. Please note that supplementary tables (should remain/ be uploaded) as separate "supporting information" files"

Reviewers' comments:

Reviewer's Responses to Questions

**Comments to the Author**

1. Is the manuscript technically sound, and do the data support the conclusions?

Reviewer #1: Partly

2. Has the statistical analysis been performed appropriately and rigorously? 

Reviewer #1: N/A

3. Have the authors made all data underlying the findings in their manuscript fully available?

Reviewer #1: Yes

4. Is the manuscript presented in an intelligible fashion and written in standard English?

Reviewer #1: Yes

5. Review Comments to the Author

Reviewer #1: Comments on “Barrier and enablers to young people accessing sexual and reproductive health services in pacific island countries and territories: a scoping review”

Dear authors,

First of all, I would like to congratulate you for all your hard works on the area of sexual and reproductive health and your effort to explore the available literatures in this topic. However, there are some rooms for improvements which can be found below. Please address these issues.

1. First of all, it is very critical to include just 8 articles for scoping review

2. Including just young people defined as 10-20 years seemed tricky to study about the sexual and reproductive health services.

3. The study has talked about actions of health care providers giving condoms and contraceptives to only married men and women affirming the perception that only married people were eligible to access family health services (result: lack of sexual and reproductive health knowledge)

- However, the study included the participants aged 10-20 years. Should young people aged, for example aged 10-15~16 years aged be given the contraceptives from the health care providers? Is the policy of PICTs support this?

- Also, the age group identified in this study mostly comprises of unmarried population (or say too early to get married), so it is unlikely to compare these results with married population.

4. The themes created to portray Barriers and Enabler of SRH in PICTs shows the strength of the study however, the articles itself contradicts the information provided in the articles;

- For example; behavior of health care provider is shown as both barriers and enabler

- Similarly, the parent knowledge and perception was also shown as both barriers and enablers

5. It is recommended to include the late adolescents (rather than 10-20 years) in the study which would actually show the barriers and enablers of SRH, since it is too early for the early adolescent to seek/access and utilize SRH (by themselves)

6. Referencing should be checked since some of the references is missing (ref. 37 and directly ref. 55 is shown > ref 38-54 is missing)

7. I am not fully authorized for the English language checking but, some grammatical corrections is recommended in some sentences.

6. PLOS authors have the option to publish the peer review history of their article (what does this mean?). If published, this will include your full peer review and any attached files.

Reviewer #1: No

---

## [Author Response · Author response to Decision Letter 0]

28 Jul 2022

The authors response to reviewer comments has been upload as a document titled: Response to reviewer

---

## [Decision Letter · Decision Letter 1]

4 Oct 2022

PONE-D-21-11951R1Barriers and enablers to young people accessing sexual and reproductive health services in Pacific Island Countries and Territories: a scoping reviewPLOS ONE

Dear Dr. Baigry,

Thank you for submitting your manuscript to PLOS ONE. After careful consideration, we feel that it has merit but does not fully meet PLOS ONE’s publication criteria as it currently stands. Therefore, we invite you to submit a revised version of the manuscript that addresses the points raised during the review process. The original referee was not available to comment on the revision, and a new reviewer has provided comments. As you will see, the reviewer is positive about the work and the revisions, and has requested minor revisions for additional information and clarification. Can you please revise the manuscript to address the issues raised?

We look forward to receiving your revised manuscript.

Kind regards,

Vanessa Carels

Staff Editor

PLOS ONE

Journal Requirements:

Reviewers' comments:

Reviewer's Responses to Questions

**Comments to the Author**

1. If the authors have adequately addressed your comments raised in a previous round of review and you feel that this manuscript is now acceptable for publication, you may indicate that here to bypass the “Comments to the Author” section, enter your conflict of interest statement in the “Confidential to Editor” section, and submit your "Accept" recommendation.

Reviewer #2: All comments have been addressed

2. Is the manuscript technically sound, and do the data support the conclusions?

Reviewer #2: Partly

3. Has the statistical analysis been performed appropriately and rigorously? 

Reviewer #2: N/A

4. Have the authors made all data underlying the findings in their manuscript fully available?

Reviewer #2: Yes

5. Is the manuscript presented in an intelligible fashion and written in standard English?

Reviewer #2: Yes

6. Review Comments to the Author

Reviewer #2: Thank you for conducting this scoping review to explore a topic which, thus far, has had little attention.

I am pleased to review R1 of this paper and can see the authors have appropriately addressed the queries raised by the reviewers of the original submission.

Abstract:

The authors revision of the abstract structure has improved the readership.

General Comments:

My advice to the authors is to publish their scoping review protocol in future, which can be done by depositing in their University's online repository or through an online repository, such as FigShare. That said, they have provided the protocol as an Appendix, which is also acceptable.

The authors only searched three databases (Medline, Scopus, CINAHL). I would recommend the inclusion of some other databases (e.g. Embase, PsychInfo) to ensure comprehensive identification of all published reports. However, it is likely they have identified the bulk of the suitable papers, given their search identified 589 unique papers.

Items to address:

1) Please state in the text what other sources were searched to identify the 40 papers included in Identification section of Figure 1. This would be best inserted after the sentence about the databases searched.

2) Please check that the 2 papers attributed to Kennedy are separate studies and are not reporting the same data from the same study (and therefore the same study population). Likewise, for the 2 papers by White. If so, you will need to address this in the limitations as it could skew the conclusion you draw from the data. This is a particularly important methodological issue for themes where the evidence is only derived from one study population, rather than reported across multiple study populations (e.g. the theme "Friendly non judgemental and kind healthcare providers" is only supported by data published by Kennedy).

3) Can the authors clarify in the manuscript whether they used original quotes from the papers, or the themes derived & reported in the papers, as the data for the thematic analysis?

4) Can the authors include the number of reviewers for each step of the data screening process, and include their initials in brackets in the text? Also, can the authors include who did the thematic analysis and include their initials in brackets in the text?

7. PLOS authors have the option to publish the peer review history of their article (what does this mean?). If published, this will include your full peer review and any attached files.

Reviewer #2: **Yes: **Dr Jacqueline Stephens

---

## [Author Response · Author response to Decision Letter 1]

1 Dec 2022

I have attached our response to the second reviews feedback in the inventory.

---

## [Decision Letter · Decision Letter 2]

6 Jan 2023

Barriers and enablers to young people accessing sexual and reproductive health services in Pacific Island Countries and Territories: a scoping review

PONE-D-21-11951R2

Dear Dr. Baigry,

We’re pleased to inform you that your manuscript has been judged scientifically suitable for publication and will be formally accepted for publication once it meets all outstanding technical requirements.

Kind regards,

Adetayo Olorunlana, Ph.D.

Academic Editor

PLOS ONE

Additional Editor Comments (optional):

Reviewers' comments:

Reviewer's Responses to Questions

**Comments to the Author**

1. If the authors have adequately addressed your comments raised in a previous round of review and you feel that this manuscript is now acceptable for publication, you may indicate that here to bypass the “Comments to the Author” section, enter your conflict of interest statement in the “Confidential to Editor” section, and submit your "Accept" recommendation.

Reviewer #2: All comments have been addressed

Reviewer #3: All comments have been addressed

2. Is the manuscript technically sound, and do the data support the conclusions?

Reviewer #2: Yes

Reviewer #3: Yes

3. Has the statistical analysis been performed appropriately and rigorously? 

Reviewer #2: Yes

Reviewer #3: N/A

4. Have the authors made all data underlying the findings in their manuscript fully available?

Reviewer #2: Yes

Reviewer #3: Yes

5. Is the manuscript presented in an intelligible fashion and written in standard English?

Reviewer #2: Yes

Reviewer #3: Yes

6. Review Comments to the Author

Reviewer #2: I am pleased to review R2 of this paper and can see the authors have appropriately addressed my queries and have made the suggested changes to the manuscript.

Reviewer #3: The authors have address previous reviewers' comments sufficiently. Although there are some minor edits to consider... in the abstract the authors used "youths" it should be just "YOUTH". There is also a word missing between "...hundred eighty..." it should be "...hundred AND eighty...". lastly, the authors need to update the page numbers on appendix S3- it is not aligned with the main text.

7. PLOS authors have the option to publish the peer review history of their article (what does this mean?). If published, this will include your full peer review and any attached files.

Reviewer #2: No

Reviewer #3: No

---

## [Editor Report · Acceptance letter]

18 Jan 2023

PONE-D-21-11951R2 

Barriers and enablers to young people accessing sexual and reproductive health services in Pacific Island Countries and Territories: a scoping review 

Dear Dr. Baigry:

I'm pleased to inform you that your manuscript has been deemed suitable for publication in PLOS ONE. Congratulations! Your manuscript is now with our production department. 

Kind regards, 

on behalf of

Associate Professor Adetayo Olorunlana 

Academic Editor

PLOS ONE